# Impact of HPV Vaccination on the Incidence of High-Grade Cervical Intraepithelial Neoplasia (CIN2+) in Women Aged 20–25 in the Northern Part of Norway: A 15-Year Study

**DOI:** 10.3390/vaccines12040421

**Published:** 2024-04-16

**Authors:** Marte Pettersen Mikalsen, Gunnar Skov Simonsen, Sveinung Wergeland Sørbye

**Affiliations:** 1Department of Medical Biology, UiT The Arctic University of Norway, 9019 Tromsø, Norway; mmi079@uit.no (M.P.M.); gunnar.skov.simonsen@unn.no (G.S.S.); 2Department of Microbiology and Infection Control, University Hospital of North Norway, 9019 Tromsø, Norway; 3Department of Clinical Pathology, University Hospital of North Norway, 9019 Tromsø, Norway

**Keywords:** HPV vaccination, cervical intraepithelial neoplasia (CIN), cervical cancer prevention, immunization programs, epidemiology of HPV, Norway, public health impact, screening programs, vaccine effectiveness, cancer epidemiology

## Abstract

Background: Human papillomavirus (HPV), the most prevalent sexually transmitted infection globally, is a key risk factor for high-grade cervical lesions and cervical cancer. Since 2009, HPV vaccination has been part of the national immunization program for girls in 7th grade in Norway (women born 1997 and later). This study aimed to assess the impact of HPV vaccination on the incidence of high-grade cervical precursors (CIN2+) among women aged 20–25 in Troms and Finnmark over a 15-year period. Materials and Methods: In this time series study, we analyzed cervical screening data from 15,328 women aged 20–25 in Troms and Finnmark, collected between 2008 and 2022. Statistical methods, including linear and logistic regression, were employed to evaluate changes in cervical intraepithelial neoplasia grade 2 and worse (CIN2+) incidence and compare risks between vaccine-offered cohorts and pre-vaccine cohorts. Results: The incidence of CIN2+ initially increased from 31 cases per year in 2008 to 110 cases in 2018, then significantly decreased to 44 cases per year by 2022 (*p* < 0.01). Women in pre-vaccine cohorts had a substantially higher risk of CIN2+ (OR 9.02, 95% CI 5.9–13.8) and CIN3+ (OR 19.6, 95% CI 7.3–52.6). Notably, no vaccinated women with CIN2+ tested positive for HPV types 16 or 18. Furthermore, none of the 13 cervical cancer cases recorded during the study were from the vaccinated cohorts. Interpretation: The findings suggest a significant reduction in the incidence of high-grade cervical precursors following the introduction of the HPV vaccine in Norway’s national immunization program, highlighting its effectiveness in cervical cancer prevention among young women in Northern Norway.

## 1. Introduction

Cervical cancer is the fourth most common cancer in women worldwide, primarily caused by persistent infection with high-risk human papillomavirus (hrHPV) [1]. Despite a global decline in its incidence and mortality, largely attributed to organized screening and HPV vaccination in developed countries, cervical cancer rates continue to escalate in less developed regions lacking these preventive measures. In Norway, a significant 40% reduction in cervical cancer has been observed since 1970 [2]. Yet, recent trends from the Cancer Registry indicate an increasing incidence in women aged 35–49 since 2007 and in those aged 25–34 since 2013. The reasons behind this rise are not entirely clear but are influenced by HPV exposure and participation in screening programs.

Cervical cancer is largely preventable, often developing over years through identifiable precancerous stages. The Norwegian Cervical Cancer Screening Program, operational since 1995, recommends regular screening for women aged 25 to 69. This program aims to detect and treat precancerous conditions before they develop into cancer. As of 2023, the program has transitioned from triennial cytology screening to HPV testing every five years, improving early detection across age groups. Positive HPV tests prompt further cytological examination of the same sample.

Globally, HPV is the most common sexually transmitted infection, with up to 80% of women and men contracting it at some point [3]. Most HPV infections are temporary and asymptomatic, being cleared by the immune system without intervention. However, persistent infection with high-risk HPV types significantly increases the risk of developing high-grade lesions and cervical cancer [4]. HPV is also implicated in other malignancies, including cancers of the oropharynx, vagina, vulva, anus, and penis [5,6,7].

In Norway, three HPV vaccines have been authorized: Cervarix, Gardasil, and Gardasil 9. Cervarix, a bivalent vaccine, targets HPV types 16 and 18, which are responsible for approximately 71% of cervical cancer cases. Gardasil, a quadrivalent vaccine, covers these two types plus types 6 and 11. Gardasil 9, a nonavalent vaccine, extends protection to five additional types (31, 33, 45, 52, and 58), covering around 91% of cervical cancer cases [8].

Cervarix has demonstrated a remarkable 93% effectiveness in preventing cervical intraepithelial neoplasia grade 3 (CIN3) lesions, irrespective of HPV type [9]. Both Cervarix and Gardasil offer cross-protection against HPV types 31, 33, and 45, likely due to the *L1* gene, which is common across these types and essential for producing the viral capsid protein that the vaccine-induced antibodies target [10]. The viral capsid protein L1 is not expressed on the surface of infected cells. Consequently, the HPV vaccine likely has no effect on the regression of pre-existing HPV infections and cellular abnormalities, and there is thus no evidence for the use of the vaccine as a therapeutic treatment [11]. Nevertheless, several studies have demonstrated a reduced risk of recurrence of high-grade lesions (CIN2+) after conization in women who have received the HPV vaccine [12].

The HPV vaccination program in Norway started in 2009 for girls in 7th grade and was extended to boys in 2018. A catch-up program was also available for women born in 1991 or later, from 2016 to 2019. The choice of vaccine for the national program is decided through a competitive tendering process conducted every four years by the Norwegian Institute of Public Health. Until the fall of 2017, Gardasil was the vaccine of choice for 12–13-year-olds in Norway, providing protection against the HPV types most commonly associated with cervical cancer and genital warts. Since then, Cervarix has been selected and used in the childhood immunization program.

Recent data from Norway’s national vaccination registry, System for Vaccination Control, (SYSVAK) highlights an encouraging trend in HPV vaccine uptake. Coverage rates for girls born between 1997 and 1999 show a progressive increase: 67% for those born in 1997, 76% for 1998, and 79% for 1999, with at least one dose of the vaccine received [13]. Since the integration of the HPV vaccine into the national childhood immunization program, vaccine coverage has notably improved. Among 7th graders, both girls and boys, the coverage rates have reached 88–90% over the past five years [13].

The primary aim of this study is to examine the trends in the incidence of high-grade cervical intraepithelial neoplasia (CIN2+) in women aged 20–25 in the Troms and Finnmark region over a 15-year period. A particular emphasis is placed on assessing the impact of introducing the HPV vaccine into the childhood immunization program. We aim to determine if this intervention has led to a decrease in high-grade precursors to cervical cancer in vaccinated cohorts. Additionally, this study seeks to evaluate the prevalence of HPV type 16, 18, and 45 using a 3-type HPV mRNA test in vaccine-offered cohorts (women born 1997 and later) and pre-vaccine cohorts.

## 2. Material and Methods

### 2.1. Summary of Methodological Objectives

This study aims to rigorously evaluate the impact of the human papillomavirus (HPV) vaccination program on the incidence of high-grade cervical intraepithelial neoplasia (CIN2+), including CIN3 and cervical cancer, among young women aged 20–25 in the Troms and Finnmark region of Norway. Over a 15-year period, encompassing the time before and after the introduction of the HPV vaccine, we systematically collected and analyzed data from the SymPathy (Tietoevry, Espoo, Finland) database, which compiles comprehensive cervical cytological and histological sample records. Our methodological approach encompasses detailed data collection, meticulous sample recording and diagnostic criteria adherence, extensive HPV mRNA testing, and robust statistical analyses.

### 2.2. Sample Recording and Diagnostic Criteria

In Norway, the utilization of unique social security numbers assigned to every citizen enables the meticulous tracking of individuals across various national health databases. This system facilitates longitudinal and retrospective studies with exceptional accuracy and minimal loss-to-follow-up. Specifically, our study leverages this infrastructure, drawing data from the following key registries:The National Cancer Registry: Mandates the reporting of all precancerous conditions and cancer cases. It serves as a central repository for cancer data, enabling comprehensive monitoring and research on cancer incidence and outcomes across the country.The National Cervical Cancer Screening Program: Collects data from all cervical cancer screenings conducted nationwide. This program ensures that screening results are systematically recorded and analyzed, contributing to public health efforts in cancer prevention.The SYSVAK National Vaccination Registry: Records all vaccinations administered, including HPV vaccinations. This registry allows researchers to track vaccination rates and evaluate the effectiveness of vaccination programs in preventing HPV-related diseases.

The integration of these registries, supported by the Scandinavian model of unique personal identification for every resident, allows for the detailed and accurate tracking of health data over time. This system is instrumental in our ability to conduct a longitudinal analysis of the impact of HPV vaccination on the incidence of CIN2+ among women in Troms and Finnmark. We recognize that such an integrated data management system is unique to Scandinavia and may not be directly comparable to systems in other countries. However, the principles of comprehensive data collection and analysis are relevant to public health research globally, offering insights into effective strategies for cancer prevention and control.

In our study, the population comprised women aged 20–25 residing in Troms and Finnmark. While the Norwegian Cervical Cancer Screening Program officially recommends screening starting at age 25, it is not uncommon for women aged 20–24 to undergo opportunistic screening. This practice is facilitated by general practitioners (GPs) during gynecological examinations for reasons such as birth control consultations or checks for sexually transmitted diseases (STDs). In these instances, cervical cytology is often performed, occasionally without a specific medical indication. Additionally, young women presenting with symptoms typically consult their GP, where cervical samples are frequently taken before any referral to a specialist in gynecology. Our study acknowledges the inclusion of such opportunistic screening data, reflecting the broader screening practices within this age group.

We employed the SymPathy database, the clinical database and laboratory information system (LIS) used at the Clinical Pathology Department at the University Hospital of North Norway. This database meticulously records all cervical cytological and histological samples from women in Troms and Finnmark counties, covering approximately 5% of Norway’s population. The SymPathy database, along with other LIS computer programs used in pathology departments across Norway, serves as a crucial component of Norway’s comprehensive approach to cervical cancer prevention, facilitating the systematic recording, analysis, and reporting of diagnostic samples. Histological diagnoses are based on well-defined criteria, with each diagnosis assigned a specific Norwegian Pathology Coding System (NORPAT) diagnostic code [14]. For identifying CIN2+ cases, we used codes M74007 (cervical intraepithelial neoplasia grade 2 (CIN2)), M80702 (cervical intraepithelial neoplasia grade 3 (CIN3)), M80703 (squamous cell carcinoma (SCC)), M81402 (adenocarcinoma in situ (ACIS)), and M81403 (adenocarcinoma (ACC)).

For women who presented with multiple cervical lesions or had undergone multiple biopsies over time, including those who had both biopsies and a conization procedure (LEEP), we prioritized the lesion with the highest grade for registration in our analysis. This approach ensured that our data accurately reflected the most severe pathological findings for each individual, aligning with our aim to assess the impact of the HPV vaccination on high-grade cervical precancerous lesions and cervical cancer.

Our study’s assessment of the incidence of CIN2+ is based on the crude numbers of women with CIN2+ identified from samples registered at the Clinical Pathology Department at the University Hospital of North Norway, which receives all cervical samples from women residing in Troms and Finnmark. This approach makes our study population-based, as our department effectively serves as the sole processing facility for these samples in the region. However, it is important to note that the calculation of CIN2+ incidence in our study does not employ the total population of women aged 20–25 years in Troms and Finnmark as the denominator, but, rather, it is derived from the number of women 20–25 years of age with CIN2+ identified among the samples received and analyzed.

We acknowledge that we did not verify the exact numbers of the total female population aged 20–25 years in the region nor the precise coverage of the cervical cancer screening program. Nevertheless, based on the national guidelines and healthcare practices, we estimate the screening program coverage to be approximately 70–80% among eligible women [15]. This estimated coverage rate, alongside our department’s comprehensive sample receipt, allows us to present a robust analysis of CIN2+ incidence trends and the impact of HPV vaccination within the studied demographic.

To confirm the vaccination status of women in vaccinated cohorts diagnosed with CIN2+, we cross-referenced our data with the SYSVAK national registry (System for Vaksinasjonskontroll), managed by the Norwegian Institute of Public Health. This step was crucial for verifying that the individuals included in our analysis had indeed received the HPV vaccine, further validating our study’s conclusions regarding the vaccine’s efficacy in reducing high-grade cervical lesion incidence.

### 2.3. HPV mRNA Testing

The assessment of HPV presence was conducted using the PreTect SEE (PreTect AS, Klokkarstua, Norway), which is a qualitative test that employs NASBA (nucleic acid sequence-based amplification) for identifying full-length E6/E7 transcripts. This method allows for the direct genotyping of the amplified messenger ribonucleic acid (mRNA) specific to HPV types 16, 18, and 45. It also incorporates an internal control within the sample to verify the adequacy of the sample collected [16].

In Norway, the cervical cancer screening program initiated in 1995 recommends screening for women aged 25–69. Historically, cervical cytology has been the primary screening method. However, the inclusion of a 14-type HPV DNA test as a primary screening tool was introduced for women aged 34–69 starting in 2019, and expanded to include women aged 25–69 from 2023 [17]. Women aged 20–24 who fall outside the primary target of the screening program are still offered cervical cytology during opportunistic screenings or if presenting symptoms. In an effort to enhance screening quality and reduce the incidence of cervical cancer following false-negative cytology results, the Department of Clinical Pathology at UNN has, since 2013, implemented the 3-type HPV mRNA test alongside cervical cytology [18,19]. Results from this test are registered in the SymPathy database, alongside all other screening outcomes, ensuring a comprehensive record of screening results.

### 2.4. Statistical Analysis

We employed descriptive statistics to calculate annual incidences of CIN2+, CIN3+, and cervical cancer from 2008 to 2022. Linear regression was used to analyze the relationship between the year and the incidence of these conditions, with separate analyses for 2008–2017 and 2017–2022. This division allowed the evaluation of the expected decrease in incidence post-2017 due to increased vaccination.

Logistic regression was utilized to assess the correlation between vaccination status and the risk of CIN2+, with odds ratios and confidence intervals computed. Chi-square tests were conducted to explore associations between vaccination status, CIN2+, CIN3+, cervical cancer, and HPV infection. Statistical analyses were performed using Statistical Package for the Social Sciences (SPSS) version 29.0 and R software version 4.3.3, with a significance threshold set at *p* < 0.05 [20]. Findings were visually represented through graphs created with SigmaPlot version 15.0 [21].

### 2.5. Categorization of Vaccination Status

For analyzing the risk of CIN2+ among vaccinated versus unvaccinated women, we classified individuals born in 1997 or later as vaccinated, and those born in 1996 or earlier as unvaccinated. Women born between 1991 and 1996, although eligible for catch-up vaccinations from 2016 to 2019, were considered unvaccinated due to low vaccine coverage and potential pre-vaccination HPV exposure.

### 2.6. Ethical Approval

The study protocol was approved as a quality assurance project by the Regional Committees for Medical and Health Research Ethics, Norway (REK Nord, ref 230825). As this study utilized anonymized data, it was not required to seek additional approvals. According to Norwegian regulations, quality assurance projects are exempt from the requirement of obtaining written informed consent from participants.

## 3. Results

### 3.1. Coverage of the HPV Vaccine

Since the HPV vaccine’s inclusion in the national immunization program for 7th-grade girls (approximately 12 years old) in 2009, vaccination coverage in Troms and Finnmark county has steadily increased. Coverage rates grew from 69.7% for women born in 1997 to 86.4% for those born in 2002 [13]. This trend indicates a significant uptake of the HPV vaccine in the northern part of Norway, aligning closely with the national coverage rates, as detailed in Table 1.

### 3.2. Trends in Incidence of High-Grade Cervical Intraepithelial Neoplasia (CIN2+, CIN3+) and Cervical Cancer in Vaccinated and Unvaccinated Cohorts

From 2008 to 2016, prior to implementation of the HPV vaccine in the childhood immunization program, the annual incidence of CIN2+ cases rose from 31 to 100 cases in Troms and Finnmark. The population at risk did not change significantly during this period. This trend peaked in 2018, with 110 cases, coinciding with vaccination of the two youngest cohorts. Following this, a significant decline was observed from 2018 to 2022, with the incidence dropping to 44 cases per year, a 60% reduction. Figure 1 illustrates this trend and includes two linear regression analyses: one showing the increase from 2008 to 2017 (*p* < 0.01) and the other the decrease from 2017 to 2022 (*p* < 0.01).

Similarly, the incidence of CIN3+ cases initially increased from 18 to 38 per year during 2008–2016. However, from 2016 to 2022, a substantial decrease to 8 cases per year was noted, marking a 79% reduction. Figure 2 presents the CIN3+ incidence trend, complemented by two linear regression analyses that confirm a statistically significant relationship between the year and CIN3+ cases (*p* < 0.01).

The incidence of cervical cancer in women under 25 in Troms and Finnmark remained consistently low over the 15-year period, with only 0 to 2 cases per year. Of the 13 cervical cancer cases recorded during this timeframe, none were from cohorts vaccinated in 7th grade.

### 3.3. HPV Type Prevalence in Vaccinated and Unvaccinated Women: Assessing the Vaccine’s Impact

In this analysis, we assessed the prevalence of HPV types 16, 18, and 45 infections among women aged 20–25 years in Troms and Finnmark from 2008 to 2022. These types were specifically targeted due to their high-risk association with cervical cancer, utilizing a 3-type HPV mRNA test for detection. Women born in 1997 and later, who were offered the HPV vaccine in school (hereafter referred to as the “vaccine-offered cohorts”), were compared with those born earlier, who did not receive the HPV vaccine as part of the school immunization program (hereafter referred to as the “pre-vaccine cohorts”).

Our findings revealed that 10.6% (284 out of 2668) of women in the pre-vaccine cohorts tested positive for HPV mRNA for types 16, 18, and/or 45, indicative of an infection with one or more of these high-risk HPV types. In contrast, among women in the vaccine-offered cohorts, only 2.6% (33 out of 1223) tested positive for any of these HPV types, showcasing a lower prevalence of infection.

When focusing specifically on HPV types 16 and 18, known to be linked to approximately 70% of cervical cancer cases, the prevalence of these types was notably lower among women in the vaccine-offered cohorts. For HPV type 16, the prevalence was 6.0% in the pre-vaccine cohorts compared to 0.8% in the vaccine-offered cohorts. Similarly, for HPV type 18, the prevalence decreased from 2.9% in the pre-vaccine cohorts to 0.9% in the vaccine-offered cohorts.

Moreover, the analysis indicated that Gardasil, which offers protection against HPV types 16, 18, 6, and 11, also provided some degree of cross-protection against HPV type 45 [22]. Among the pre-vaccine cohorts, 2.7% tested positive for HPV type 45, whereas this figure was reduced to 1.4% among the vaccine-offered cohorts.

Chi-square tests highlighted statistically significant associations between vaccination status and overall HPV infection (*p* < 0.001), as well as specific associations with HPV genotypes 16 (*p* < 0.001), 18 (*p* < 0.001), and 45 (*p* = 0.01).

### 3.4. Comparative Analysis of High-Grade Cervical Lesion Risk in Vaccinated Versus Unvaccinated Women

Between 2008 and 2022, a total of 15,328 women in Troms and Finnmark underwent cervical sampling, with 934 being diagnosed with CIN2+. Logistic regression was used to explore the relationship between vaccination status and the risk of CIN2+. In this analysis, women born in 1997 or later were considered vaccinated, while those born in 1996 or earlier were classified as unvaccinated. The findings revealed a significantly higher risk of CIN2+ in unvaccinated women, with an odds ratio (OR) of 9.02 (95% CI 5.9–13.8). An even stronger association was observed for CIN3+ risk, with an OR of 19.6 (95% CI 7.3–52.6).

Among the 934 women with CIN2+, only 2.4% belonged to a vaccinated cohort (22 out of 934), compared to 97.6% who did not receive the vaccine as part of the vaccination program (912 out of 934). Similarly, of the 379 women with CIN3+, only 1.1% were part of the cohort offered vaccination (4 out of 379). Chi-square tests confirmed a statistically significant link between vaccination status and the incidence of both CIN2+ (*p* < 0.001) and CIN3+ (*p* < 0.001). Notably, none of the vaccinated women with CIN2+ tested positive for HPV types 16 or 18, which are targeted by the Gardasil vaccine.

### 3.5. Conization Procedures at UNN: A Reflection of CIN2+ Management (2008–2022)

The number of conization procedures in women 20–25 years of age analyzed at our department mirrored the trend in CIN2+ incidence, reaching a peak of 79 in 2017. However, from 2017 to 2022, there was a notable decline in conizations, averaging 21 cases per year (Figure 3). A corresponding analysis compared the frequency of conization procedures with the occurrence of women with CIN2+, revealing a decreasing trend in the proportion of lesions treated with conization. This proportion decreased from 65% in 2018 to 47% in 2022.

## 4. Discussion

This study assessed the impact of the HPV vaccine, introduced into the childhood immunization program in 2009, on the incidence of CIN2+, CIN3+, and cervical cancer among young women in Troms and Finnmark over 15 years. Notably, from 2018 to 2022, we observed a significant 60.0% reduction in CIN2+ incidence (*p* < 0.01) and a 79% decrease in CIN3+ incidence (*p* < 0.01) from 2016 to 2022. Among women diagnosed with CIN2+, only 2.4% were vaccinated, and none of these vaccinated women were infected with HPV types 16 or 18, targeted by the Gardasil vaccine. Furthermore, no cervical cancer cases were recorded among the cohorts offered the HPV vaccine in 7th grade during the entire study period (2008–2022). Additionally, there was a notable disparity in HPV mRNA positivity (HPV 16, 18, and 45) between vaccinated and unvaccinated women with normal cytology (2.7% vs. 10.9%, *p* < 0.001).

The observed decrease in CIN2+ incidence from 2017 to 2022, coinciding with the increasing proportion of vaccinated cohorts, underlines the impact of the HPV vaccination program. It is important to consider the progression time for CIN2+ development in our analysis. Our study spans a 15-year period (from 2008 to 2022), initially including a population of 20–25-year-old women, none of whom were vaccinated (born before 1997). As the study progressed, an increasing proportion of women in this age group belonged to cohorts that had been offered the HPV vaccine. By the end of the study period, all women in this age group were part of vaccinated cohorts. This transition from an entirely unvaccinated to a fully vaccinated population within our study’s age range provides a unique perspective. The reduction in CIN2+ incidence observed is most likely attributable to the vaccination, rather than the natural progression time for developing CIN2+. The comprehensive nature of this study, encompassing the period both before and after the vaccine’s introduction, inherently considers the typical timeline for CIN2+ development. This is reflected in the contrasting trends observed in the pre-vaccination and post-vaccination periods. Such an analysis strengthens our understanding of the vaccine’s efficacy in real-world settings and its pivotal role in reducing the incidence of high-grade cervical lesions.

In evaluating the impact of HPV vaccination on the incidence of CIN2+ and CIN3+ among women aged 20–25 in Troms and Finnmark, we meticulously considered age to be a crucial variable. This specific age range was chosen because it represents a pivotal transition period for individuals who were among the first cohorts eligible for HPV vaccination under Norway’s national immunization program [23]. As these individuals move from late adolescence into early adulthood, they embody a group at significant risk for HPV exposure and the development of related cervical pathologies. The selection of this age range allowed us to assess the direct effects of the HPV vaccination before the onset of regular cervical cancer screening recommended at age 25. It is essential to understand that while age itself did not interfere with our analysis, it was an integral factor in framing the context of our study. The consistent age range across the 15-year study period facilitated a clear, longitudinal assessment of the vaccine’s impact, minimizing confounding variables and enabling a focused examination of trends in CIN2+ and CIN3+ incidences. This approach ensured that any observed changes in the incidence rates could be more confidently attributed to the effects of vaccination rather than other demographic or behavioral factors that might influence HPV infection risks. By focusing on this age group, our findings offer valuable insights into the early benefits of vaccination and underscore the importance of early vaccination efforts in reducing the prevalence of high-grade cervical lesions, supporting the vaccine’s role in preemptive cervical cancer prevention strategies.

Our findings of an increasing CIN2+ incidence from 2008 to 2016 correlate with Orumaa et al.’s Norwegian study [24], which reported escalating incidence rates of CIN2, CIN3, and AIS (adenocarcinoma in situ) from 1992 to 2016. This rise is attributed to changes in screening protocols and diagnostic practices, including the adoption of liquid-based cytology and HPV testing, which improved detection rates. Additionally, factors like early sexual debut and multiple sexual partners increased exposure to sexually transmitted infections, including HPV. This is supported by the National Institute of Public Health’s 2016 report, which showed a rise in chlamydia and gonorrhea cases during this period [25]. Moreover, studies by Serrano et al. indicate an increase in other HPV-related cancers [26]. The concurrent rise in sexually transmitted infections and HPV-related cancers suggests an elevated exposure to HPV, contributing to the increased CIN2+ cases observed in our study.

The decline in CIN2+ incidence from 2017 to 2022 corresponds with the increasing proportion of vaccinated cohorts in our study population. This reduction was anticipated, given the vaccination program’s expansion. However, it is notable that CIN2+ incidence peaked in 2018 with the inclusion of the two youngest cohorts. This delay in peak incidence could be due to several factors:Initial lower vaccine coverage: The first vaccinated cohort (girls born in 1997) had relatively lower vaccine coverage compared to later years.Higher HPV exposure in older women: Women at the upper end of our age range likely had greater exposure to HPV and a higher risk of cellular changes, impacting the overall trend.Time lag between HPV infection and cellular changes: The progression from HPV infection to cellular changes can span several years, possibly explaining the lower incidence of CIN2+ among 20-year-olds compared to 25-year-olds.

Differences in HPV exposure and CIN2+ incidence across age groups were likely consistent throughout the study period, playing a minor role in the observed temporal trends.

Our findings of a marked reduction in CIN2+ cases, particularly among vaccinated women, are in line with the existing literature on HPV vaccine efficacy. For instance, Feiring et al. [27] reported a significant decrease in HPV prevalence among Norwegian 17-year-olds from the 1997 cohort compared with unvaccinated girls born in 1994, showing a 42% overall reduction in HPV infection and an 81% reduction for HPV types targeted by Gardasil. This study also highlighted the benefits of herd immunity and cross-protection, with both vaccinated and unvaccinated women in the 1997 cohort showing reduced HPV prevalence.

In considering the potential disparities in the number of samples between vaccinated and unvaccinated women in relation to HPV mRNA positivity, it is essential to acknowledge the context of Norway’s HPV vaccination program. The program, which is school-based, has consistently achieved high coverage rates among 12-year-old girls, increasing from approximately 70% in 2009 to 90% by 2022 [13]. Such widespread uptake mitigates the likelihood of significant sampling bias between vaccinated and unvaccinated populations in our study. Concerns have been raised about whether HPV-vaccinated women might exhibit lower attendance at adult screening programs compared to their unvaccinated counterparts. However, recent research indicates that HPV-vaccinated women attend cervical screening at rates comparable to those who are unvaccinated, suggesting that vaccination status does not adversely affect screening participation [28]. Furthermore, our laboratory’s implementation of the HPV mRNA test as part of its internal quality assurance process—applied uniformly across all screening tests—ensures that our analysis is not skewed by differential screening behaviors. This uniform application, aimed at reducing the risk of cervical cancer following normal cytology, allows for an equitable comparison of HPV mRNA positivity rates between the two groups. While women testing positive for HPV receive more rigorous follow-up, this procedure is standard across both vaccinated and unvaccinated individuals, ensuring that our findings reflect the genuine impact of HPV vaccination on mRNA positivity rates without bias.

Our study’s comparison to Kjaer et al.’s 2020 study [29], which investigated high-grade cell changes in vaccinated women aged 16–23 in Denmark, Iceland, Norway, and Sweden, reveals some important distinctions. Kjaer et al. observed no high-grade cell changes related to HPV types 16 or 18 among vaccinated women after 12 years. In contrast, our study encompassed both vaccinated and unvaccinated women and used CIN2+ of all HPV types as the endpoint, not just those associated with HPV types 16 and 18. Consequently, while we also observed no cases of CIN2+ among vaccinated women linked to these HPV types, our broader focus across all HPV genotypes offered a more comprehensive view of CIN2+ incidence post-vaccination. This indicates that, despite differences in study design and endpoints, the HPV vaccine’s role in reducing precursors to cervical cancer is consistent across multiple studies. This trend mirrors the progress observed in international studies, such as the recent comprehensive analysis from Scotland, which reported no cases of invasive cervical cancer among women vaccinated at ages 12 or 13, underscoring the profound protective effects of timely HPV vaccination [30].

In Troms and Finnmark, the reduction in conizations from 79 in 2017 to 21 in 2022 paralleled the decline in CIN2+ incidence. This decrease is likely due to fewer precancerous lesions and a shift towards regular monitoring instead of immediate conization, considering the high rate of lesion regression in younger women [31]. This trend aligns with the Cancer Registry’s 2021 report, which noted a decline in conizations among Norwegian women under 25, from 323 in 2017 to 122 in 2022 [32]. These findings reinforce the effectiveness of the HPV vaccination program in reducing both the incidence of high-grade cervical lesions and the need for invasive procedures like conizations.

In reflecting upon our findings, it is crucial to contextualize the screening practices for women aged 20–24 in light of national recommendations and recent data [33]. While our study period observes women in this age range, recent health statistics from 2022 indicate no cases of cervical cancer in Norway among women aged 25 years or younger, who belong to the vaccinated cohorts [15]. This suggests that the HPV vaccination has played a significant role in reducing the incidence of cervical cancer to zero in this demographic [30]. Furthermore, it highlights an important consideration regarding the treatment of CIN2 or CIN3 in women younger than 25 years of age, which, in light of these outcomes, could be reconsidered to avoid potential overtreatment [34,35].

Given these considerations, the practice of opportunistic screening in women aged 20–24, while previously thought to offer early detection benefits, must now be weighed against the backdrop of highly effective HPV vaccination coverage and its impact on the incidence of high-grade lesions and cervical cancer in this population. Our study’s insights into the negligible prevalence of CIN2+ in vaccinated cohorts underline the effectiveness of the HPV vaccination program and invite a reevaluation of screening strategies for young women in Norway and similar contexts. This evolving landscape offers an opportunity to align screening practices more closely with the evidenced reduction in risk, potentially modifying the approach to managing CIN2+ and CIN3 lesions in younger women.

## 5. Strengths and Limitations

### 5.1. Strengths


Extensive observation period: The 15-year span of this study allows for a thorough analysis of trends pre- and post-HPV vaccine introduction, reducing the impact of random variations and enhancing the reliability of findings.High-quality data source: Data from the SymPathy database, covering all cervical cytological and histological samples from Troms and Finnmark, ensures precision and minimizes information bias.Rigorous diagnostic process: The use of two independent senior physicians for high-grade biopsy assessments at UNN, supplemented by p16 immunostaining, bolsters diagnostic accuracy.Digital diagnostic tools: Incorporating machine learning and AI in evaluating CIN2+ biopsies has improved diagnostic reproducibility.Real-world data utilization: This approach provides a realistic view of clinical practices and enhances the generalizability of the findings, reflecting the real-world target population more accurately than traditional clinical trials.


### 5.2. Limitations


Lack of control group: The absence of a control group limits the ability to establish a direct causal relationship between HPV vaccination and observed trends.Potential selection bias: Although the study includes all women with cervical samples in the region, those who did not participate in screening programs are not represented, which could introduce selection bias.Regional specificity: The findings are specific to Troms and Finnmark, and extrapolating results to different geographical or demographic contexts requires caution.Vaccination status uncertainties: The study assumes vaccination status based on birth year, which do not accurately reflect individual vaccination histories.Potential confounding factors: Other factors influencing HPV prevalence and CIN2+ incidence, such as changes in sexual behavior or screening practices, are not controlled for in the study.Opportunistic screening in younger age group: Women aged 20–24 years are not included in the Norwegian organized cervical cancer screening program, and all CIN2+ cases detected in this age group result from opportunistic screening. This context may affect the comparability and interpretation of CIN2+ incidence among younger women, underscoring the need for cautious evaluation of these findings within the broader screening and vaccination landscape.


## 6. Conclusions

This study has demonstrated a significant shift in the incidence of CIN2+ among young women in Troms and Finnmark between 2008 and 2022. Initially, from 2008 to 2017, an increase in CIN2+ incidence was observed, likely due to changes in the screening program and increased HPV exposure. However, following the integration of the HPV vaccine into the childhood immunization program, a notable decrease in CIN2+ incidence occurred from 2017 to 2022. This reduction coincided with a higher proportion of the study population being vaccinated. Importantly, women in pre-vaccine cohorts (born before 1997) demonstrated a significantly higher risk of developing both CIN2+ and CIN3+ lesions compared to women in cohorts offered the HPV vaccine (born 1997 and later). Furthermore, none of the vaccinated women with CIN2+ were infected with HPV types 16 or 18, the genotypes targeted by the Gardasil vaccine. Additionally, all 13 cases of cervical cancer recorded during the study period occurred in women who were not part of the vaccinated cohorts.

These findings corroborate the effectiveness of the HPV vaccine in significantly reducing the incidence of high-grade cervical precursors. They provide strong evidence supporting the public health benefit of including the HPV vaccine in childhood immunization programs, aligning with international research underscoring the vaccine’s critical role in cervical cancer prevention.

## Figures and Tables

**Figure 1 vaccines-12-00421-f001:**
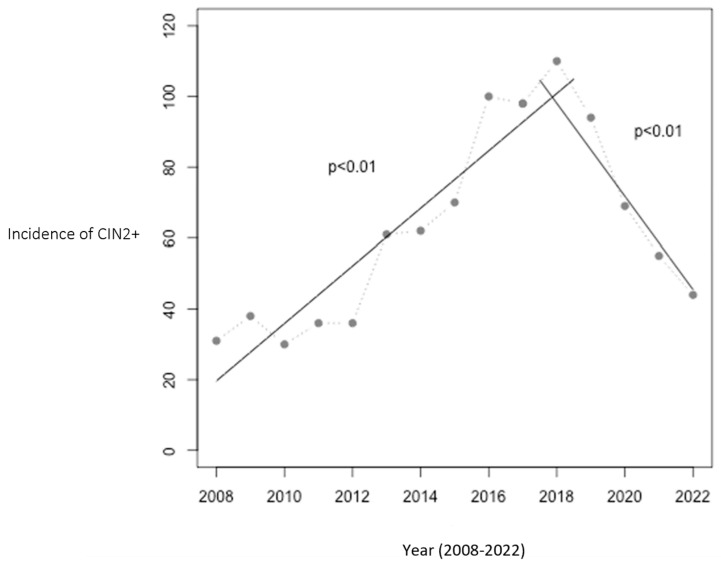
Number of women aged 20–25 with CIN2+ in Troms and Finnmark. The relationship between year and the incidence of CIN2+ was statistically significant with *p* < 0.01 in both periods (2008–2017 and 2017–2022). The HPV vaccination program was initiated in 2009, with the first cohort of vaccinated girls becoming 20 years old in 2017 and reaching 25 years of age in 2022, marking the study’s observation period.

**Figure 2 vaccines-12-00421-f002:**
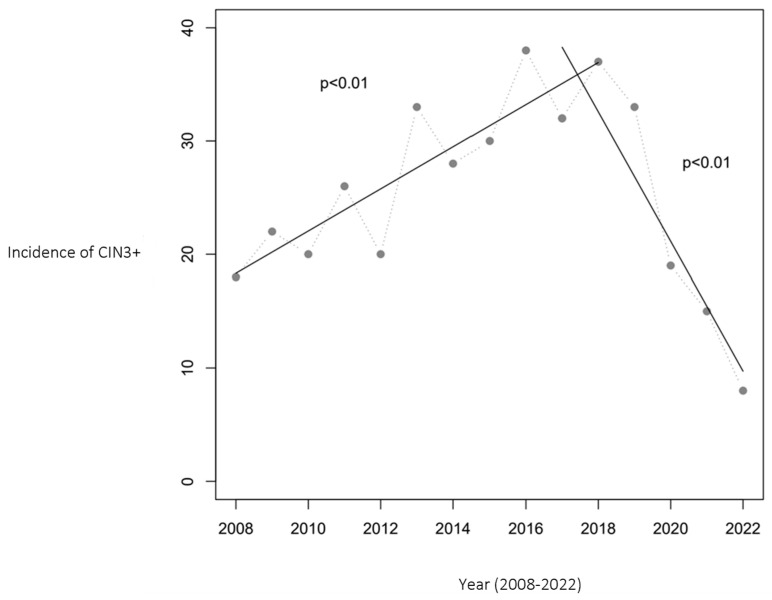
Number of women aged 20–25 with CIN3+ in Troms and Finnmark. The relationship between year and the incidence of CIN3+ was statistically significant, with *p* < 0.01 in both periods (2008–2017 and 2017–2022). The HPV vaccination program was initiated in 2009, with the first cohort of vaccinated girls becoming 20 years old in 2017 and reaching 25 years of age in 2022, marking the study’s observation period.

**Figure 3 vaccines-12-00421-f003:**
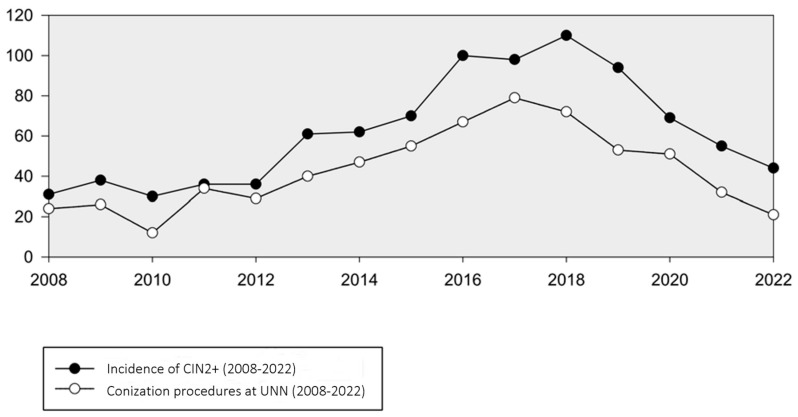
Incidence of CIN2+ and number of conization procedures in women aged 20–25 years at UNN (2008–2022).

**Table 1 vaccines-12-00421-t001:** Comparison of HPV vaccination coverage rates in Troms and Finnmark County versus Norway (1997–2002).

Birth Year	Troms and Finnmark	Norway
1997	69.7	67.1
1998	76.8	76.3
1999	80.3	78.7
2000	80.5	81.1
2001	88.3	88.1
2002	86.4	88.3

## Data Availability

The raw data supporting the conclusions of this article will be made available by the authors on request.

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
