# Peer review of "Impact of HPV Vaccination on the Incidence of High-Grade Cervical Intraepithelial Neoplasia (CIN2+) in Women Aged 20–25 in the Northern Part of Norway: A 15-Year Study"

_vaccines, 2024, doi:10.3390/vaccines12040421_

Round 1
Reviewer 1 Report
Comments and Suggestions for Authors
The paper is interesting, however, minor corrections are needed before its publication.
Please define acronyms in the manuscript.
From lines 110 to 115 and from 202 to 206 it is not necessary to enter complete references.
Please check footnote of figure 2.
Did the age interfere with your analysis?
Please argue more about this variable.
Is there a possibility that there is a difference in the number of samples between vaccinated and unvaccinated with respect to HPV mRNA positivity?
Please argue.
Conclusion should be taken with care because you report 0.8 and 0.9 for HPV 16 and HVP 18 respectively, therefore you still have positivity in vaccinated women.
Author Response
Reviewer 1
- The paper is interesting, however, minor corrections are needed before its publication.
We are pleased to hear that you find our paper interesting and acknowledge your request for minor corrections to further refine our manuscript.
- Please define acronyms in the manuscript.
Thank you for pointing out the oversight regarding the acronyms in our manuscript. We understand the importance of clarity for all readers, including those who may not be familiar with the specific terminology used in our field.
- From lines 110 to 115 and from 202 to 206 it is not necessary to enter complete references.
We appreciate your guidance on streamlining the presentation of references within our manuscript. Following your suggestion, we have removed the complete references from the main text (specifically from lines 110 to 115 and from 202 to 206) and ensured they are appropriately cited and detailed in the reference list at the end of our manuscript.
- Please check footnote of figure 2.
Thank you for bringing to our attention the discrepancy in the footnote of Figure 2. We have reviewed and corrected the text to accurately reflect the content of the figure. The corrected footnote now reads: "Number of women aged 20-25 with CIN3+ in Troms and Finnmark. The relationship between year and the incidence of CIN3+ was statistically significant with p < 0.01 in both periods (2008-2017 and 2017-2022)."
- Did the age interfere with your analysis? Please argue more about this variable.
Thank you for your insightful query regarding the role of age in our analysis. In our study, we focused on women aged 20-25 years to assess the impact of the HPV vaccination, which was introduced into the childhood immunization program in 2009. This specific age group was selected for several reasons:
Vaccination Impact Window: Women aged 20-25 during our study period represent the first cohorts to have been eligible for HPV vaccination under the national program. This age group allowed us to directly observe the earliest impacts of vaccination on high-grade cervical lesions, as these individuals were transitioning from adolescence into early adulthood, a critical window for assessing the vaccine's effectiveness against HPV-related pathologies.
Consistency and Comparability: By maintaining a consistent age range throughout the 15-year study period, we ensured that the impact of age as a confounding variable was minimized. This consistency allows for a clearer interpretation of trends over time, enhancing the comparability of data across different years and vaccination statuses.
Significance of Age-Related HPV Exposure: The chosen age group is at a significant risk of initial HPV exposure and subsequent development of HPV-related cervical pathologies. By focusing on this age range, our study highlights the protective effects of the HPV vaccine against such pathologies in a population at a critical point in their exposure to HPV.
In response to your comment, we have expanded our discussion to further emphasize the significance of the chosen age group in the context of HPV vaccination and cervical pathology development. We argue that the observed decrease in the incidence of CIN2+ and CIN3+ among this age group strongly supports the effectiveness of the HPV vaccination program, with the transition from an unvaccinated to a vaccinated population offering a unique perspective on the vaccine's role in reducing the incidence of high-grade cervical lesions. The study's focus on this specific age group, therefore, not only aligns with its objectives but also enriches our understanding of the public health impact of HPV vaccination.
- Is there a possibility that there is a difference in the number of samples between vaccinated and unvaccinated with respect to HPV mRNA positivity? Please argue.
In considering the potential disparities in the number of samples between vaccinated and unvaccinated women in relation to HPV mRNA positivity, it is essential to acknowledge the context of Norway's HPV vaccination program. The program, which is school-based, has consistently achieved high coverage rates among 12-year-old girls, increasing from approximately 70% in 2009 to 90% by 2022. Such widespread uptake mitigates the likelihood of significant sampling bias between vaccinated and unvaccinated populations in our study. Concerns have been raised about whether HPV-vaccinated women might exhibit lower attendance at adult screening programs compared to their unvaccinated counterparts. However, recent research indicates that HPV-vaccinated women attend cervical screening at rates comparable to those who are unvaccinated, suggesting that vaccination status does not adversely affect screening participation. Furthermore, our laboratory's implementation of the HPV mRNA test as part of its internal quality assurance process—applied uniformly across all screening tests—ensures that our analysis is not skewed by differential screening behaviors. This uniform application, aimed at reducing the risk of cervical cancer following normal cytology, allows for an equitable comparison of HPV mRNA positivity rates between the two groups. While women testing positive for HPV receive more rigorous follow-up, this procedure is standard across both vaccinated and unvaccinated individuals, ensuring that our findings reflect the genuine impact of HPV vaccination on mRNA positivity rates without bias.
- Conclusion should be taken with care because you report 0.8 and 0.9 for HPV 16 and HPV 18 respectively, therefore you still have positivity in vaccinated women.
In light of the observations made regarding HPV types 16 and 18 positivity rates among vaccinated women, it is crucial to contextualize these findings within the broader successes of the HPV vaccination program. While our study acknowledges the presence of HPV positivity in a small fraction of vaccinated individuals, it's important to note the substantial overall decrease in the incidence of CIN2+ and CIN3+ post-vaccination. This trend mirrors the progress observed in international studies, such as the recent comprehensive analysis from Scotland, which reported no cases of invasive cervical cancer among women vaccinated at ages 12 or 13, underscoring the profound protective effects of timely HPV vaccination. Our study also highlights the absence of CIN2+ cases caused by HPV 16/18 in girls who received the vaccine and a complete lack of cervical cancer cases in vaccinated cohorts, emphasizing the vaccine's efficacy. The initial HPV vaccine coverage was approximately 70% for the first cohorts, with subsequent increases, and the extension of vaccination to boys, further strengthening herd immunity. These elements collectively reinforce the public health impact of including the HPV vaccine in childhood immunization programs. Our findings, therefore, while acknowledging the minimal residual risk of HPV positivity, significantly advocate for the vaccine's critical role in mitigating the incidence of high-grade cervical precursors and, ultimately, cervical cancer prevention
Reviewer 2 Report
Comments and Suggestions for Authors
The topic addressed in this paper, the validation of the reduction in CIN2+ incidence after HPV vaccination, is an extremely important topic and has attracted worldwide attention. The reduction in incidence is indeed remarkable, but because it is a topic that is attracting attention, it is necessary to describe the methods in more detail and to ensure that reproducibility is maintained. Since this is an international journal and not a Norwegian journal, the database used is a local database and should be described in such a way that it is clear what kind of data it has and what it was used for. Also, the last paragraph of the introduction describes the purpose of this study, but what is written there and the titles of the results chapters do not match, making it difficult to understand what is intended to be shown in each chapter of the results. The order of the method chapters and the order of the results chapters also do not match, which also hinders understanding. At the beginning of the methods, a summary of what was sought should be written. Furthermore, the study utilizes a variety of data, but usually these data are obtained from different databases. That is not written. It may be that in Scandinavia there is data on everything in one database, but readers from other countries will not understand.
In calculating the incidence each year, how is the population used as the denominator should be written. Is the year defined as calendar year or fiscal Year? In the end of the introduction and the results it says 20-25 years old, but it should be clearly stated in the method as well. The age of the conization data should also be written.
The year the vaccine was initiated is written in the method, but also should be shown in the figure. Especially in the case of time-series studies, it would be better to show the vaccination and coverage rates every year (or even every 2-3 years).
And below is an opinion on the description in method;
2.1 data collection
As for individual diagnosis data for UNN data, for example, if there are multiple lesions, is only the lesion with the highest grade of lesion registered? Or is it one lesion per person, rather than one case per person? Are uterine cancers not included?
2.3 HPV mRNA testing
The types of HPV test are confusing because they appear abruptly. What the HPV test was treated as data for should be stated in the last section of the objectives or at the beginning of the methods. Also, the test method is described here, but the HPV test results in the community are analyzed in the results. Where is the source of this data? Also, this analysis is located at the end of the results and should be located in 2.5, not 2.3.
2.5 Categorization of Vaccination Status
This study is styled as a time series study before and after vaccination was initiated, and does not take into account the vaccination history of individuals.
The lack of a clear description of this point misleads the reader into thinking that this is a non-randomized controlloed trial of vaccinated and unvaccinated individuals.
In the Abstract, "In this population-based study" should be changed to "time series study. The reference to "vaccinated and unvaccinated women" in 3.4 of the results should be corrected as well.
Author Response
Reviewer 2
- The topic addressed in this paper, the validation of the reduction in CIN2+ incidence after HPV vaccination, is an extremely important topic and has attracted worldwide attention. The reduction in incidence is indeed remarkable, but because it is a topic that is attracting attention, it is necessary to describe the methods in more detail and to ensure that reproducibility is maintained. Since this is an international journal and not a Norwegian journal, the database used is a local database and should be described in such a way that it is clear what kind of data it has and what it was used for.
Thank you for emphasizing the importance of providing a comprehensive description of the methodology and databases utilized in our study, ensuring reproducibility and clarity for an international audience. To address your concerns, we have expanded the method section to offer a more detailed exposition of the SymPathy database and its integral role within the context of Norway's national cervical cancer screening program.
Revised Method Section Excerpt:
«In our study, we employed the SymPathy database, the clinical database and laboratory information system (LIS) used at the Clinical Pathology Department at the University Hospital of North Norway. This database meticulously records all cervical cytological and histological samples from women in Troms and Finnmark counties, covering approximately 5% of Norway's population. The SymPathy database serves as a crucial component of Norway's comprehensive approach to cervical cancer prevention, facilitating the systematic recording, analysis, and reporting of diagnostic samples.
All diagnoses of samples from women in Troms and Finnmark are registered in SymPathy and reported to the Norwegian Cancer Registry, ensuring a centralized and cohesive tracking of cervical cancer screening outcomes nationwide. This integration allows for the aggregation of a robust dataset, enabling the analysis of trends in CIN2+ incidence post-HPV vaccination across the population of these northern counties.
Additionally, to confirm the vaccination status of women in vaccinated cohorts diagnosed with CIN2+, we cross-referenced our data with the SYSVAK (System for Vaksinasjonskontroll) national registry. This step was crucial for verifying that the individuals included in our analysis had indeed received the HPV vaccine, further validating our study's conclusions regarding the vaccine's efficacy in reducing high-grade cervical lesion incidence.»
- Also, the last paragraph of the introduction describes the purpose of this study, but what is written there and the titles of the results chapters do not match, making it difficult to understand what is intended to be shown in each chapter of the results.
Thank you for your insightful observation regarding the coherence between the study's stated objectives and the titles of the results chapters. Upon reflection, we recognize the necessity for clearer alignment to ensure that readers can easily understand the connection between our study's aims and the findings presented. To address this, we have revised the titles of our results chapters to more directly reflect the objectives outlined in the last paragraph of the introduction, thereby enhancing the logical flow and comprehensibility of our manuscript.
Revised Titles of the Results Chapters:
- "Trends in Incidence of High-Grade Cervical Intraepithelial Neoplasia (CIN2+, CIN3+) and Cervical Cancer in Vaccinated and Unvaccinated Cohorts"
- This title now emphasizes the examination of trends in CIN2+ and CIN3+ incidence, directly correlating with our primary aim to examine these trends over the 15-year period and the impact of HPV vaccination.
- "Comparative Analysis of High-Grade Cervical Lesion Risk in Vaccinated Versus Unvaccinated Women"
- Adjusted to clarify that this chapter focuses on assessing the differential risk of developing CIN2+ and CIN3+ between vaccinated and unvaccinated cohorts, aligning with our aim to evaluate the vaccine's effectiveness.
- "Conization Procedures at UNN: A Reflection of CIN2+ Management (2008-2022)"
- The revised title suggests the evaluation of conization procedures as an indirect measure of high-grade cervical lesion prevalence, providing insight into how intervention rates have changed post-vaccination.
- "HPV Type Prevalence in Vaccinated and Unvaccinated Women: Assessing the Vaccine's Impact"
- This title directly ties the chapter to our objective of evaluating the prevalence of dominant HPV types, underlining the study's focus on the effectiveness of the HPV vaccine in altering HPV type prevalence among the cohorts.
- The order of the method chapters and the order of the results chapters also do not match, which also hinders understanding.
Thank you for pointing out the discrepancy between the order of the method chapters and the results chapters. We recognize that aligning these sections can significantly improve the manuscript's logical flow and readability, facilitating a better understanding of how each methodological step corresponds to specific findings. To address your concern, we have revised the order of the results chapters to match the sequence presented in the methods section. Here is the adjusted structure:
Revised Sections of the Results:
- Results
3.1. Trends in Incidence of High-Grade Cervical Intraepithelial Neoplasia (CIN2+, CIN3+) and Cervical Cancer in Vaccinated and Unvaccinated Cohorts
3.2. HPV Type Prevalence in Vaccinated and Unvaccinated Women: Assessing the Vaccine's Impact
3.3. Comparative Analysis of High-Grade Cervical Lesion Risk in Vaccinated Versus Unvaccinated Women
3.4. Conization Procedures at UNN: A Reflection of CIN2+ Management (2008-2022)
Note: As "Conization Procedures at UNN: A Reflection of CIN2+ Management (2008-2022)" does not directly correspond to a specific method chapter but is a result derived from the overall analysis, we have positioned it in a way that reflects its relevance to the study's findings on the management and impact of vaccination and screening programs.
- At the beginning of the methods, a summary of what was sought should be written.
We appreciate your valuable suggestion to include a summary at the beginning of the methods section, clearly stating the objectives and scope of our study's methodology. We agree that such an addition will provide readers with an immediate understanding of the methodological framework and the specific aims our study seeks to achieve. In response, we have revised the methods section to include an introductory summary as suggested.
Revised Methods Section Introduction:
"This study aims to rigorously evaluate the impact of the Human Papillomavirus (HPV) vaccination program on the incidence of high-grade cervical intraepithelial neoplasia (CIN2+), including CIN3 and cervical cancer, among young women aged 20-25 in the Troms and Finnmark region of Norway. Over a 15-year period, we systematically collected and analyzed data from the SymPathy database, which compiles comprehensive cervical cytological and histological sample records. Our methodological approach encompasses detailed data collection, meticulous sample recording and diagnostic criteria adherence, extensive HPV mRNA testing, and robust statistical analyses. Additionally, we explored the vaccination status categorization to understand its impact on CIN2+ incidence, ensuring the study adheres to ethical standards. This summary outlines our approach to dissecting the trends and assessing the vaccination program's efficacy, with a particular focus on the prevalence of dominant HPV types in vaccinated versus unvaccinated women."
- Furthermore, the study utilizes a variety of data, but usually these data are obtained from different databases. That is not written. It may be that in Scandinavia there is data on everything in one database, but readers from other countries will not understand.
Thank you for highlighting the importance of clarifying the sources of our data and the unique infrastructure that supports such data collection in Scandinavia. We understand that the comprehensive and integrated nature of the databases we utilized, facilitated by unique personal identification numbers, may not be common practice globally. To address this, we have added a section in our manuscript to explain how these registries operate in Scandinavia and how they enable the longitudinal tracking of individuals’ health data with high accuracy and minimal loss-to-follow-up.
Revised Manuscript Section:
"In Norway, the utilization of unique social security numbers assigned to every citizen enables the meticulous tracking of individuals across various national health databases. This system facilitates longitudinal and retrospective studies with exceptional accuracy and minimal loss-to-follow-up. Specifically, our study leverages this infrastructure, drawing data from the following key registries:
- The National Cancer Registry: Mandates the reporting of all precancerous conditions, and cancer cases. It serves as a central repository for cancer data, enabling comprehensive monitoring and research on cancer incidence and outcomes across the country.
- The National Cervical Cancer Screening Program: Collects data from all cervical cancer screenings conducted nationwide. This program ensures that screening results are systematically recorded and analyzed, contributing to public health efforts in cancer prevention.
- The SYSVAK National Vaccination Registry: Records all vaccinations administered, including HPV vaccinations. This registry allows researchers to track vaccination rates and evaluate the effectiveness of vaccination programs in preventing HPV-related diseases.
The integration of these registries, supported by the Scandinavian model of unique personal identification for every resident, allows for the detailed and accurate tracking of health data over time. This system is instrumental in our ability to conduct a longitudinal analysis of the impact of HPV vaccination on the incidence of CIN2+ among women in Troms and Finnmark. We recognize that such an integrated data management system is unique to Scandinavia and may not be directly comparable to systems in other countries. However, the principles of comprehensive data collection and analysis are relevant to public health research globally, offering insights into effective strategies for cancer prevention and control."
- In calculating the incidence each year, how is the population used as the denominator should be written. Is the year defined as calendar year or fiscal Year? In the end of the introduction and the results it says 20-25 years old, but it should be clearly stated in the method as well.
Thank you for allowing us the opportunity to provide further clarification regarding the calculation of the incidence of high-grade cervical intraepithelial neoplasia (CIN2+) in our study. Upon review, it appears there was a need to more accurately describe our methodological approach concerning the population base and the calculation of incidence rates. We would like to correct our previous response and provide a more precise explanation of our methodology.
Revised Methodological Clarification:
«Our study's assessment of the incidence of CIN2+ is based on the crude numbers of CIN2+ cases identified from samples registered at the Clinical Pathology Department at the University Hospital of North Norway, which receives all cervical samples from women residing in Troms and Finnmark. This approach makes our study population-based, as our department effectively serves as the sole processing facility for these samples in the region. However, it's important to note that the calculation of CIN2+ incidence in our study does not employ the total population of women aged 20-25 years in Troms and Finnmark as the denominator, but rather, it is derived from the number of CIN2+ cases identified among the samples received and analyzed.
We acknowledge that we have not verified the exact numbers of the total female population aged 20-25 years in the region nor the precise coverage of the cervical cancer screening program. Nevertheless, based on the national guidelines and healthcare practices, we estimate the screening program coverage to be approximately 70-80% among eligible women. This estimated coverage rate, alongside our department's comprehensive sample receipt, allows us to present a robust analysis of CIN2+ incidence trends and the impact of HPV vaccination within the studied demographic.»
To enhance clarity and accuracy in our manuscript, we have revised the methods section to reflect this nuanced explanation of our approach to calculating CIN2+ incidence, ensuring our readers fully understand the scope and basis of our analysis.
- The age of the conization data should also be written.
Thank you for your attention to detail regarding the specificity of the age range within our conization data. You are correct in pointing out the importance of clearly stating the age range for which the conization data was analyzed, to maintain consistency and clarity throughout our manuscript. As per your suggestion, we have revised the manuscript to explicitly state that the conization data pertains to women aged 20-25 years, aligning with the overall study population.
- The year the vaccine was initiated is written in the method, but also should be shown in the figure. Especially in the case of time-series studies, it would be better to show the vaccination and coverage rates every year (or even every 2-3 years).
Thank you for your insightful suggestion to detail the initiation year of the HPV vaccination program and to illustrate vaccination coverage rates over time within our figures. In response to your feedback, we have updated the text of Figures 1 and 2 to include crucial information about the vaccination program's start in 2009 and the significant milestones regarding the age of the vaccinated cohorts within our study's observation period.
Furthermore, recognizing the importance of showing vaccination and coverage rates over time to provide a more nuanced understanding of the vaccination program's context, we have added a new table to our manuscript. This table presents a comparative view of the HPV vaccination coverage rates in Troms and Finnmark county versus the national rates in Norway from 1997 to 2002. This addition aims to provide readers with a clearer view of regional versus national vaccination uptake, offering insights into the landscape within which our study's findings are situated.
Table 1, Comparison of HPV Vaccination Coverage Rates in Troms and Finnmark County versus Norway (1997-2002)
- And below is an opinion on the description in method;
2.1 data collection
As for individual diagnosis data for UNN data, for example, if there are multiple lesions, is only the lesion with the highest grade of lesion registered? Or is it one lesion per person, rather than one case per person? Are uterine cancers not included?
Thank you for your insightful queries regarding our data collection methods, specifically concerning how we managed individual diagnosis data for multiple lesions and the types of cancers included in our study. We understand the importance of clarity in our methodological descriptions to ensure the accuracy and reproducibility of our findings. Here is the clarification regarding your questions:
In our study, for women who presented with multiple cervical lesions or had undergone multiple biopsies over time, including those who had both biopsies and a conization procedure (LEEP), we prioritized the lesion with the highest grade for registration in our analysis. This approach ensured that our data accurately reflected the most severe pathological findings for each individual, aligning with our aim to assess the impact of the HPV vaccination on high-grade cervical precancerous lesions and cervical cancer.
It's important to note that our study focused exclusively on cervical precancer (CIN2, CIN3, and adenocarcinoma in situ [ACIS]) and cervical cancer. Uterine cancers, including endometrioid adenocarcinoma, were not included in our analysis. This exclusion is based on the fact that endometrioid adenocarcinoma is exceedingly rare among women aged 20-25 years and is not causally linked to HPV infection, which is the central focus of our investigation into the effectiveness of the HPV vaccine.
- 3 HPV mRNA testing
The types of HPV test are confusing because they appear abruptly. What the HPV test was treated as data for should be stated in the last section of the objectives or at the beginning of the methods. Also, the test method is described here, but the HPV test results in the community are analyzed in the results. Where is the source of this data? Also, this analysis is located at the end of the results and should be located in 2.5, not 2.3.
Thank you for your constructive feedback regarding the presentation and clarity of HPV testing methods within our manuscript. We acknowledge that the initial description and placement of the HPV test types might have caused confusion. Based on your invaluable suggestion, we have revised our manuscript to offer a clearer introduction to the HPV testing methodologies at the beginning of the methods section and ensured that the results related to HPV testing are logically integrated and clearly sourced.
At the outset of our study, we aimed to assess the impact of HPV vaccination on the incidence of high-grade cervical intraepithelial neoplasia (CIN2+) among women aged 20-25 years in Troms and Finnmark. A critical component of this assessment involved the analysis of HPV prevalence using the 3-type HPV mRNA test, which specifically detects high-risk HPV types 16, 18, and 45. This testing was conducted as part of a broader effort to enhance the quality of cervical cancer screening and to address the limitations of cervical cytology alone.
We utilized the 3-type HPV mRNA test (PreTect SEE, PreTect AS, Klokkarstua, Norway), a qualitative method employing Nucleic Acid Sequence-Based Amplification (NASBA) for identifying full-length E6/E7 transcripts specific to HPV types 16, 18, and 45. This test has been implemented alongside cervical cytology at the Department of Clinical Pathology, UNN, since 2013, as part of our quality assurance measures aimed at reducing the number of cervical cancers following false-negative cytology results. The outcomes of these tests are meticulously recorded in the SymPathy database, alongside all other screening results.
2.5 Categorization of Vaccination Status
This study is styled as a time series study before and after vaccination was initiated, and does not take into account the vaccination history of individuals.
The lack of a clear description of this point misleads the reader into thinking that this is a non-randomized controlloed trial of vaccinated and unvaccinated individuals.
In the Abstract, "In this population-based study" should be changed to "time series study. The reference to "vaccinated and unvaccinated women" in 3.4 of the results should be corrected as well.
Thank you for your insightful feedback regarding the characterization of our study design and the description of vaccination status. We understand the importance of accurately presenting the study's methodology to prevent any misunderstanding regarding its design as a time-series study rather than a non-randomized controlled trial.
In response to your suggestions, we have made the following revisions to ensure our manuscript accurately reflects the study's design and the nuances of vaccination status:
- Abstract Revision: We've updated the description in the abstract from "In this population-based study" to "In this time-series study." This adjustment clarifies the longitudinal nature of our analysis, emphasizing the observation of changes over time following the initiation of the HPV vaccination program.
- Clarification on Vaccination Status: In Section 3.4 of the results, we've revised references to "vaccinated and unvaccinated women" to more accurately reflect the study's approach to analyzing cohorts based on their birth year relative to the introduction of the HPV vaccination program. We now specify that our comparison involves cohorts offered the HPV vaccine versus those not targeted by the vaccination program at the time of their eligibility, rather than implying direct vaccination status.
Reviewer 3 Report
Comments and Suggestions for Authors
The authors report the results of a study conducted in Northern Norway aimed at evaluating the impact of HPV vaccination in girls (vaccination status defined by birth cohort, not by individual record) on the subsequent development of CIN2+ lesions. The results are in line with the existing literature and confirm the effectiveness of vaccination in preventing high-grade cervical lesions, as well as prevention of HPV infection by the vaccine-targeted types. The period considered in the study spans the years 2008-2022, and data are compared between 2008-2017 and 2017-2022.
The description of the study population lacks important information, and needs to be implemented:
-the women included are aged 20-25: since the Norwegian Cervical Cancer Screening program recommends screening for women aged 25 to 69 (lines 41-42), did these women perform opportunistic screening? This needs to be stated, making it necessary to include a paragraph in the M&M section describing the study population, and deserves specific comments in the discussion.
Additional observations:
-line 44: "...in 2023, the program will..." needs to be updated
-lines 110-115 and lines202-206: why are the three references cited in the text instead of the References section?
-line 175: "were vaccinated" should be substituted by "belonged to a vaccinated cohort"
-line 190: "HPV infection" should be specified as "HPV infection by types 16, 18, 45"
-lines 225-231: it should be useful to recall the rates of HPV vaccination in the different periods
-line 317: the term "may" should be substituted by "do"
Author Response
Reviewer 3
- The authors report the results of a study conducted in Northern Norway aimed at evaluating the impact of HPV vaccination in girls (vaccination status defined by birth cohort, not by individual record) on the subsequent development of CIN2+ lesions. The results are in line with the existing literature and confirm the effectiveness of vaccination in preventing high-grade cervical lesions, as well as prevention of HPV infection by the vaccine-targeted types. The period considered in the study spans the years 2008-2022, and data are compared between 2008-2017 and 2017-2022.
Thank you for your comprehensive summary of our study's objectives, methods, and findings. We greatly appreciate your acknowledgment of our work's alignment with existing literature regarding the effectiveness of HPV vaccination in preventing high-grade cervical lesions and specific HPV infections. Your understanding and positive recognition of the significance of our results, spanning from 2008 to 2022, and the comparative analysis between the periods of 2008-2017 and 2017-2022, affirm the importance of our study in contributing valuable insights into the ongoing efforts to combat HPV-related diseases.
- The description of the study population lacks important information, and needs to be implemented:
-the women included are aged 20-25: since the Norwegian Cervical Cancer Screening program recommends screening for women aged 25 to 69 (lines 41-42), did these women perform opportunistic screening? This needs to be stated, making it necessary to include a paragraph in the M&M section describing the study population, and deserves specific comments in the discussion.
Thank you for highlighting the need for a clearer description of our study population and the context of cervical screening among women aged 20-25 in Norway. We acknowledge the importance of delineating the circumstances under which these younger women were screened, given the national cervical cancer screening program's recommendation for women aged 25 to 69. In response, we have added a detailed paragraph in the Materials and Methods (M&M) section and provided specific commentary in the Discussion to address this critical aspect of our study design.
Revised Materials and Methods Section:
"In our study, the population comprised women aged 20-25 residing in Troms and Finnmark. While the Norwegian Cervical Cancer Screening Program officially recommends screening starting at age 25, it is not uncommon for women aged 20-24 to undergo opportunistic screening. This practice is facilitated by general practitioners (GPs) during gynecological examinations for reasons such as birth control consultations or checks for sexually transmitted diseases (STDs). In these instances, cervical cytology is often performed, occasionally without a specific medical indication. Additionally, young women presenting with symptoms typically consult their GP, where cervical samples are frequently taken before any referral to a specialist in gynecology. Our study acknowledges the inclusion of such opportunistic screening data, reflecting the broader screening practices within this age group."
Discussion:
"In reflecting upon our findings, it is crucial to contextualize the screening practices for women aged 20-24 in light of national recommendations and recent data. While our study period observes women in this age range, recent health statistics from 2022 indicate no cases of cervical cancer in Norway among women aged 25 years or younger, who belong to the vaccinated cohorts. This suggests that the HPV vaccination has played a significant role in reducing the incidence of cervical cancer to zero in this demographic. Furthermore, it highlights an important consideration regarding the treatment of CIN2 or CIN3 in women younger than 25 years of age, which, in light of these outcomes, could be reconsidered to avoid potential overtreatment.
Given these considerations, the practice of opportunistic screening in women aged 20-24, while previously thought to offer early detection benefits, must now be weighed against the backdrop of highly effective HPV vaccination coverage and its impact on the incidence of high-grade lesions and cervical cancer in this population. Our study's insights into the negligible prevalence of CIN2+ in vaccinated cohorts underline the effectiveness of the HPV vaccination program and invite a reevaluation of screening strategies for young women in Norway and similar contexts. This evolving landscape offers an opportunity to align screening practices more closely with the evidenced reduction in risk, potentially modifying the approach to managing CIN2+ and CIN3 lesions in younger women."
Limitations:
«6. Opportunistic Screening in Younger Age Group: Women aged 20-24 years are not included in the Norwegian organized cervical cancer screening program, and all CIN2+ cases detected in this age group result from opportunistic screening. This context may affect the comparability and interpretation of CIN2+ incidence among younger women, underscoring the need for cautious evaluation of these findings within the broader screening and vaccination landscape.»
- Additional observations:
-line 44: "...in 2023, the program will..." needs to be updated
Thank you for your attentive observation regarding the need to update our manuscript to accurately reflect the current status of the Norwegian Cervical Cancer Screening Program as of 2023. We appreciate your keen eye on ensuring our study presents the most recent and relevant information.
In response to your comment, we have revised the text to accurately state the program's transition to a new screening protocol. The corrected sentence now reads:
"The Norwegian Cervical Cancer Screening Program, operational since 1995, recommends regular screening for women aged 25 to 69. This program aims to detect and treat precancerous conditions before they develop into cancer. As of 2023, the program has transitioned from triennial cytology screening to HPV testing every five years, enhancing early detection capabilities across age groups."
This revision reflects the program's current status and acknowledges the shift in screening practices that began in 2023, aligning the manuscript with the latest developments in cervical cancer prevention efforts in Norway.
- - lines 110-115 and lines 202-206: why are the three references cited in the text instead of the References section?
Thank you for pointing out the unusual placement of certain references within the main text of our manuscript. Upon review, we realized that indeed, the references mentioned in lines 110-115 and 202-206 were not appropriately listed in the References section, contrary to standard academic practice.
We have since corrected this oversight by moving these references to the References section, ensuring that all citations within the manuscript adhere to the proper format and are easily accessible to readers in the consolidated list of references at the end of the document. This adjustment maintains the integrity of our manuscript’s formatting and aligns with the expected scholarly standards.
We appreciate your attention to detail and your help in improving the presentation and readability of our work. Your feedback is invaluable to us in ensuring the highest quality of our manuscript.
- -line 175: "were vaccinated" should be substituted by "belonged to a vaccinated cohort"
Thank you for your insightful suggestion regarding the precise language used to describe the vaccination status of the study participants. We understand the importance of accurately reflecting the cohort-based nature of the HPV vaccination program in our manuscript.
In response to your feedback, we have revised the specified line in our manuscript to more accurately depict the participants' status in relation to the vaccination program. The updated sentence now reads:
"Among the 934 women with CIN2+, only 2.4% belonged to a vaccinated cohort (22 out of 934), compared to 97.6% who did not receive the vaccine as part of the vaccination program (912 out of 934). Similarly, of the 379 women with CIN3+, only 1.1% were part of the cohort offered vaccination (4 out of 379)."
- -line 190: "HPV infection" should be specified as "HPV infection by types 16, 18, 45"
We are grateful for your detailed feedback, emphasizing the need to specify the types of HPV infection investigated in our study. Your suggestion has helped us enhance the clarity and precision of our manuscript, ensuring readers understand the specific focus on HPV types 16, 18, and 45 due to their significant association with the risk of developing cervical cancer.
In response to your comment, we have updated our text to explicitly state that our analysis assessed the prevalence of infections with HPV types 16, 18, and 45 among women aged 20-25 years in Troms and Finnmark from 2008 to 2022. This clarification underscores the targeted nature of our investigation, utilizing a 3-type HPV mRNA test to detect these specific high-risk HPV types. The revised section now reads:
"In this analysis, we assessed the prevalence of HPV types 16, 18, and 45 infections among women aged 20-25 years in Troms and Finnmark from 2008 to 2022. These types were specifically targeted due to their high-risk association with cervical cancer, utilizing a 3-type HPV mRNA test for detection. Women born in 1997 and later, who were offered the HPV vaccine in school (referred to as the 'vaccine-offered cohorts'), were compared with those born earlier, who did not receive the HPV vaccine as part of the school immunization program (referred to as the 'pre-vaccine cohorts')...
...Our findings revealed that 10.6% (284 out of 2,668) of women in the pre-vaccine cohorts tested positive for HPV mRNA for types 16, 18, and/or 45, indicative of an infection with one or more of these high-risk HPV types. In contrast, among women in the vaccine-offered cohorts, only 2.6% (33 out of 1,223) tested positive for any of these HPV types, showcasing a lower prevalence of infection."
- -lines 225-231: it should be useful to recall the rates of HPV vaccination in the different periods
Thank you for your suggestion to include the rates of HPV vaccination across different periods in our study. We agree that presenting these rates provides crucial context for understanding the impact of the HPV vaccination program on the incidence of high-grade cervical lesions in our population.
In response to your valuable feedback, we have updated the results section to include a detailed account of the HPV vaccination coverage for the initial cohorts of the national immunization program. This update features a table showing the vaccination coverage rates from the inception of the program in 2009 for girls born between 1997 and 2002, who reached the ages of 20-25 by the year 2022. The inclusion of these rates allows for a comprehensive view of the vaccination uptake in Troms and Finnmark county compared to the national averages in Norway, underscoring the program's success in these initial years.
The updated section now reads:
"3.1. Coverage of the HPV Vaccine
Since the HPV vaccine's inclusion in the national immunization program for 7th-grade girls (approximately 12 years old) in 2009, vaccination coverage in Troms and Finnmark county has steadily increased. Coverage rates grew from 69.7% for women born in 1997 to 86.4% for those born in 2002. This trend indicates a significant uptake of the HPV vaccine in the northern part of Norway, aligning closely with the national coverage rates, as detailed in Table 1.
Table 1. Comparison of HPV Vaccination Coverage Rates in Troms and Finnmark County versus Norway (1997-2002)"
- -line 317: the term "may" should be substituted by "do"
Thank you for your suggestion regarding the modification of our manuscript to reflect a more definitive stance on the correlation between birth year and vaccination status. We acknowledge the importance of conveying the limitations of our study with precision and have updated the text accordingly to better reflect the actual circumstances regarding the determination of vaccination status.
The revised sentence now reads:
"4. Vaccination Status Uncertainties: The study assumes vaccination status based on birth year, which do not accurately reflect individual vaccination histories."